# Initial Outcomes of the Safe and Sound Protocol on Patients with Adult Autism Spectrum Disorder: Exploratory Pilot Study

**DOI:** 10.3390/ijerph20064862

**Published:** 2023-03-09

**Authors:** Hiroki Kawai, Makiko Kishimoto, Yuko Okahisa, Shinji Sakamoto, Seishi Terada, Manabu Takaki

**Affiliations:** 1Department of Neuropsychiatry, Graduate School of Medicine, Dentistry and Pharmaceutical Sciences, Okayama University, Okayama 700-8530, Japan; celeg182003@gmail.com (H.K.);; 2National Center for Child Health and Development, Tokyo 157-0074, Japan

**Keywords:** autism, adults, auditory problems, listening therapy

## Abstract

Sensory impairments are common features of autism spectrum disorder (ASD) and are associated with its social impairments. However, there is no established treatment for these impairments in adults with ASD. The Safe & Sound Protocol (SSP) is a listening program designed to improve social communication skills by reducing auditory hypersensitivity. We investigated the effectiveness of the SSP for adults with ASD. We administered the SSP to six participants with ASD aged 21–44 years old, and the effects were assessed using the Social Responsiveness Scale, Second Edition (SRS-2). Secondary outcomes were assessed using the Center for Epidemiological Studies Depression Scale (CES-D), State-Trait Anxiety Inventory (STAI), WHO Quality of Life 26 (WHOQOL-BREF), and Adolescent/Adult Sensory Profile (A/ASP). In this study, only the Social Awareness scale of the SRS-2 Family-Report showed a significant improvement after the intervention. In addition, it was significantly correlated with physical health of WHOQOL-BREF (r = −0.577, *p* = 0.012), state and trait anxiety of STAI (r = 0.576, *p* = 0.012; r = 0.708, *p* = 0.00009, respectively), and CES-D (r = 0.465, *p* = 0.05). In conclusion, the SSP has a partial effect on social impairments in adults with ASD, specifically on the Social Awareness subscale of the SRS-2.

## 1. Introduction

Autism spectrum disorder (ASD) is a neurodevelopmental disorder characterized by impaired social ability, as well as restricted and repetitive sensory-motor behaviors. The estimated prevalence of ASD in developed countries is approximately 1.5% [1] and may be increased by the inclusion of adults undiagnosed in childhood [2]. There is limited evidence on effective treatment for adolescents and adults with ASD [3] due to comorbid developmental and psychiatric conditions [4,5]. Psychological treatments are frequently administered to individuals with ASD, but treatment efficacy varies due to differences in individual developmental status, timing of treatment initiation, and outcome measures [6,7,8]. Additionally, there are no approved pharmacological treatments for social impairment in adolescents and adults with ASD [9]. 

The polyvagal theory, which was first proposed by Porges in 1995, provides plausible explanations for the social impairment of ASD [10,11]. In summary, this hypothesis argues that a certain balance of the autonomic nervous systems (ANS) between the sympathetic and parasympathetic nervous systems (SNS and PNS, respectively) is important for appropriate social behavior in mammals, including humans. In a mammalian nervous system, unconscious decisions are continuously made by processing sensory information obtained from the environment. This unconscious neural mechanism associated with sensory processing is named neuroception, and its dysfunction hinders the emergence of appropriate social behaviors. Neuroception has a significant influence on efficiently switching between defensive behaviors such as fight, flight, or freeze to prosocial behaviors for interpersonal communication. However, individuals with ASD may have neuroceptional dysfunctions due to sensory processing abnormalities and, therefore, may not be able to switch defensive behaviors effectively due to a lack of perception of the environment as safe enough for social interaction with others. 

The Safe & Sound Protocol (SSP) is a creative intervention based on the polyvagal theory for improving social communication impairments by reducing auditory hypersensitivity and improving human speech processing (Associate Manual Safe & Sound Protocol). The SSP rehabilitates the middle ear muscle function using filtered music that is tuned to the specific frequency of human speech. Previous studies of children with ASD using an early version of SSP reported significant improvements in their sensory problems, including auditory processing, listening, and hearing sensitivities [12,13,14].

Patients with ASD experience social difficulties including expressing and receiving both nonverbal and verbal communication, as well as emotional expression. Adults with ASD tend to experience auditory sensory overload, specifically, and are frustrated by certain frequencies, loud noises, and mixtures of competing sounds [15,16]. These previous studies suggest the need for interventions to reduce auditory distortion and its related symptoms in adults with ASD. However, there are no established interventions for these issues [17]; moreover, most of the developed auditory integration training or listening therapies are designed to target pediatric ASD [18] because it is considered to be effective in younger patients [19,20]; moreover, as synaptic turnover decreases with age, plastic changes may be limited in adults [21].

Several studies on listening therapy based on the Polyvagal Theory for children with ASD have reported improvements in auditory function and social communication [13,22]; however, there are no studies of its effects on adults and adolescents with ASD. The purpose of this study is to evaluate the efficacy of SSP on social communication, auditory hypersensitivity, and psychiatric symptoms in adults with ASD, as well as its safety, feasibility, and applicability.

## 2. Materials and Methods

### 2.1. Participants

The study was conducted between December 2017 and August 2018 at the Department of Neuropsychiatry, Okayama University hospital. This study was approved by the Ethics Committee of Okayama University hospital (Rin1801-004). Six participants with ASD aged 21–44 years (average age: 27.1 years) were recruited from the outpatient Department of Neuropsychiatry of Okayama University hospital. All participants were Japanese and diagnosed with ASD based on the ADOS-2 (Autism Diagnostic Observation Schedule, Second Edition). A trained, research-licensed clinical psychiatrist performed all diagnostic assessments based on ADOS-2 criteria [23,24] or direct observation. Before the study, all participants provided informed written consent. They were not compensated.

Age, marital status, employment status, education, country of birth, and medical history were collected in face-to-face interviews to confirm that all participants met the inclusion and exclusion criteria. 

### 2.2. Inclusion, Exclusion, and Discontinuance Criteria

The inclusion criteria were: (1) aged between 20 and 50 years, (2) being naive to listening therapy, and (3) diagnosed with ASD based on the ADOS-2.

The exclusion criteria were having (1) a diagnosis of schizophrenia or bipolar disorder, (2) serious neurological or physical conditions, (3) neurodevelopmental disorders with known genetic etiology, (4) history of epilepsy, (5) a previous diagnosis of post-traumatic stress disorder, and (6) having undergone intensive cognitive behavior therapy within the last 6 months. Patients receiving medication were not excluded; however, we did exclude patients whose drug dosage and type were changed during the intervention and (7) those with severe motor, vision hearing, or chronic health problems. No participant had a history of substance use. Although this study did not include an assessment of intellectual function, no participant was considered to have obvious intellectual disabilities during the general examination and medical interview.

The discontinuation criterion was based on a score of 2 or higher on the ninth item of the Patient Health Questionnaire-9 (PHQ-9) “Thoughts that you would be better off dead or of hurting yourself in some way” at the time of the interview during the intervention.

### 2.3. Instrumentation and Testing Environment

The intervention comprised five daily sessions of 60 min each. The processed music programs were delivered through an MP3 ear-cup type headphone. The participants were allowed to adjust the correct volume for themselves from 48 to 75 decibels relative to the carrier (dBC). In the filtered music condition, vocal music was computer-processed based on a proprietary algorithm developed for removing low and high frequencies and for modulating the frequency bandwidth associated with the human voice from 50 Hz to 3000 Hz. These features characterize a mother singing a lullaby. The SSP playlist consisted mostly of folk music sung by females for adults that had been considered for tone, mood, melody, and lyrics. However, because music and people’s reactions are personal, if a certain music evoked a reaction and hindered the performance of SSP, it was discussed with the participant. Each participant listened to the SSP program on the first and last day at the outpatient clinic of Okayama University hospital, and they listened to the SSP program on the second, third, and fourth days at home. The participants were instructed on how to use SSP equipment. Moreover, during the intervention, the practitioner helped the client feel safe and comfortable to facilitate receptiveness to new acoustic stimuli and maximize the SSP effectiveness [25].

### 2.4. Assessment & Measurements

The evaluation was conducted at three time points: before and one hour after the intervention and approximately one month (28–35 days) after the intervention as a follow-up (endpoint). After the intervention and at a follow-up point, the participants and their families reported anecdotally to the researchers their impressions of using SSP. We used common ASD diagnostic, cognitive, adaptive communication, mental status, and problem behavior measures, including the Social Responsiveness Scale, Second Edition (SRS-2); Center for Epidemiological Studies Depression Scale (CES-D); State-Trait Anxiety Inventory (STAI); WHO Quality of Life 26 (WHOQOL-BREF); and Adolescent/Adult Sensory Profile (A/ASP). Details of each outcome are provided below.

1.SRS-2 Adult Self-Report and Family-Report Forms (main outcome):

The SRS-2 Adult Self-Report and Family-Report Forms contain 65 items that identify ASD-related social impairments and quantify their severity [26]. The response options range from 0 to 3 for each item, with a higher score indicating greater severity. The SRS-2 is a valid measure of autistic symptomatology across cultures [27,28]; moreover, it has a conceptually derived three-factor structure that is consistent with the DSM-5 criteria for ASD. Its factors are social communication impairment, restricted interests, and repetitive behaviors.

2.Center for Epidemiological Studies Depression Scale (CES-D):

The CES-D [29] is a 20-item measure that assesses symptoms associated with depression, including restless sleep, poor appetite, and feeling lonely. The response options range from 0 to 3 for each item. Total scores range from 0 to 60, with higher scores indicating more severe depressive symptoms. Moreover, the CES-D provides cutoff scores (e.g., ≥16) that aid in identifying individuals at risk for clinical depression with good sensitivity, specificity, and high internal consistency [30].

3.State-Trait Anxiety Inventory (STAI):

The STAI is a commonly used and reliable measure of state anxiety (A-state) and trait anxiety (A-trait) [31]. Each subscale is comprised of 20 items. All the items are rated on a 4-point scale. The total score obtained from each subscale ranges from 20 to 80, with a high score indicating a high anxiety level.

4.WHO Quality of Life 26 (WHOQOL-BREF):

The WHOQOL-BREF, which is an abbreviated 26-item version of the WHOQOL-100, was developed as a valid and reliable alternative assessment scale using data from the field-trial version of the WHOQOL-BREF-100 [32]. Individual items were rated on a 5-point scale, with each ranging from the highest to lowest score (5–1). Scores of 1 and 5 indicate the lowest negative and highest positive perceptions, respectively. The questionnaire score ranges from 26 to 130. The first question generally evaluates QOL, while the second question evaluates health condition satisfaction. The other 24 questions were grouped into 4 domains: psychological (6 items), social (3 items), environmental (8 items), and physical domains (7 items).

5.Adolescent/Adult Sensory Profile (A/ASP):

The A/ASP is a valid and reliable tool for assessing behavioral responses to sensory occurrences similar to daily life experiences developed by Brown and Dunn [33]. This 60-item self-report questionnaire is divided into six different categories: auditory, visual, smell/taste, touch, movement, and activity level. The evaluation identifies the sensory profile and provides the following four quadrant scores: Low Registration (poor sensory registration), Sensation Seeking, Sensory Sensitivity, and Sensation Avoiding that correspond to the quadrant scores provided by the Sensory Profile (SP).

### 2.5. Data Analysis

All data were statistically analyzed using the Wilcoxon signed-rank test (*p* < 0.05) because the normality of the score distribution was not clear due to the small sample size, and the Shapiro-Wilk test also showed no significant normality. The correlation between SRS-2 subscales and other secondary outcomes was evaluated with Spearman’s rank correlation coefficient (*p* < 0.05). We conducted these analyses using RStudio (an open-source statistical package) and IBM SPSS Statistics software version 19.0. 

## 3. Results

The SSP was successfully administered to all participants according to the protocol. Some participants presented mild side effects, including headaches, sleeplessness, and fatigue, but these side effects were all temporary, and no participant met the discontinuation criteria. Table 1 shows the backgrounds of all participants, including sex, age, occupation, history of psychiatric disorders, and relationships with the evaluators of SRS-2 Family-Report. There were three males and three females (average age: 27.2 ± 8.1 years old). Four participants were diagnosed with adjustment disorder; one was taking low-dose quetiapine for ASD-associated irritability.

### 3.1. Scores of SRS-2 Adult Self-Report and Family-Report Forms after the Safe & Sound Protocol

Table 2 presents the scores of the SRS-2 Self- and Family-Reports before and after the intervention, as well as the follow-up. The Social Awareness subscale of the SRS-2 Family-Report only showed significant differences for before the intervention vs. at the endpoint (*p* = 0.027, 95% CI = [6.9, 10.3]). However, the SRS-2 Self-Report showed no significant difference in any of the subscales. The other secondary outcomes also showed no significant differences after the intervention including the endpoint.

### 3.2. Correlation between Score of SRS-2 Adult Self-Report and Family-Report Forms and Secondary Outcomes

Table 3 shows the correlations between the SRS-2 Self- or Family-Reports and A/ASP. The total score of the SRS-2 Self-Report showed a positive correlation with Low Registration (r = 0.487, *p* = 0.04) and negative correlation with Sensation Seeking of A/ASP (r = −0.572, *p* = 0.01312). On the other hand, the total score of the SRS-2 Family-Report showed a significant correlation only for Low Registration (r = 0.470, *p* = 0.04853). Furthermore, Low Registration of A/ASP was significantly correlated with three SRS-2 Self- and Family-Reports subscales. Sensory Sensitivity and Sensation Avoiding of A/ASP were not correlated with the total score of either the SRS-2 Self- or Family-Reports. Additionally, we analyzed the correlation between the total score of the SRS-2 Self- or Family-Reports and other secondary outcomes. A significant correlation was observed between the total score of the SRS-2 Self-Report and the physical health (r = −0.504, *p* = 0.03289) and Environment (r = −0.542, *p* = 0.02) of WHOQOL-BREF subscales, and the state anxiety (r = 0.576, *p* = 0.01228) of the STAI subscale. In addition, the total scores of the SRS-2 Family-Report and the social relationships subscale (r = 0.714, *p* = 0.0008443) and overall WHOQOL-BREF score (r = 0.548, *p* = 0.01853) were also significantly correlated. Other than the above, secondary outcomes were not correlated with the total score of the SRS-2 Self- or Family-Report.

### 3.3. Correlation Analysis between Social Awareness Subscale of SRS-2 Adult Self-Report and Family-Report Forms and Secondary Outcomes

The Social Awareness subscale of the SRS-2 Self-Report was significantly correlated with the physical health (r = −0.541, *p* = 0.02) and environment (r = −0.6244, *p* = 0.0056) subscales of WHOQOL-BREF; state anxiety (r = 0.614, *p* = 0.0066) was correlated with STAI subscale. The Social Awareness subscale of the SRS-2 Family-Report, which was improved by SSP, was significantly correlated with the physical health subscale of the WHOQOL-BREF (r = −0.577, *p* = 0.012) (Figure 1A), state anxiety (r = 0.576, *p* = 0.012) (Figure 1B), and trait anxiety (r = 0.708, *p* = 0.00099) (Figure 1C) STAI subscales, and the CES-D (r = 0.465, *p* = 0.05) (Figure 1D). Other than the above, secondary outcomes were not correlated with the Social Awareness subscale of the SRS-2 Self-Report.

## 4. Discussion

This is the first study to conduct polyvagal theory-based listening therapy in adults with ASD. Only Social Awareness of the SRS-2 Family-Report showed improvement after the SSP intervention. Social Awareness refers to the ability to pick up on social cues and sensory aspects of reciprocal social behavior [26]. Therefore, our results suggest that the SSP may locally affect the sensory aspects of interpersonal communication in adults with ASD. On the other hand, no improvement was observed in the rest of the SRS-2 subscales, which indicate active and motor aspects of social skills.

Social difficulties are associated with sensory abnormalities in ASD [34,35]. In this study, there was no significant improvement in A/ASP before and after the intervention, but there was an association between the SRS-2 Self-Report total score and Low Registration and Sensation Seeking scores suggesting sensory hypo-responsiveness, among the four quadrants of A/ASP. A previous study of children with ASD reported by Hilton et al. showing the association between ASD symptom severity and sensory processing impairments used an assessment battery similar to ours, the original Sensory Responsiveness Scale (SRS) and the SP; they reported that Social Awareness of the SRS correlated only with hyper-responsiveness subscales of the SP such as Sensory Sensitivity and Sensation Avoiding [36]. In our results, the SRS-2 Family-Report total score was associated only with Low Registration of the A/ASP, and the SRS-2 Self-Report total score was associated with Sensation Seeking of A/ASP in addition to Low Registration suggesting sensory hypo-responsiveness, in contrast to the study of children. Although not readily comparable to reports of ASD in children, which are primarily assessed by their parents or others, it would be natural for the results not to be entirely consistent between the family-reported social impairments and the self-rated sensory impairments in adults with ASD, but it can also be hopefully interpreted as a slight easing of the defensive behavior as participants become more attentive to their environment. In addition, in a previous study of adults with ASD, Crane et al. reported that 94.4% of their ASD sample had some sensory processing impairments within the four quadrants of the A/ASP and showed a significant heterogeneity in patterns of sensory processing dysfunction across ASD groups [37]. Differences in ASD severity by age have also been reported [38,39]. These factors are likely to cause differences between children and adults in the evaluation of the effectiveness of SSP treatment.

Although there was some variation on the evaluation scale that was difficult to interpret, each participant showed unique post-intervention responses. One case (Case #1) had psychological conflicts within the family one month after the SSP intervention that required an emotional release session and family counseling. As a result, the conflicts were resolved three months after the intervention, and the participant’s adaptation to daily life had improved. In this case, the SSP seemed to guide the emergence of an unresolved internal conflict accompanied with irritability and aggression that could be managed with pharmacotherapy and psychological therapy. In another case (Case #2), the family reported an improvement in the participant’s responsiveness; however, the participant did not report any subjective change. Another participant (Case #5) was older than the other participants and reported an increase in her awareness and self-regulation of hypersensitivity itself rather than an improvement in auditory sensitivity. Finally, a participant who had suffered from nocturnal insomnia since childhood slept through the night without awakening after the SSP intervention. Moreover, the family reported improvements in self-control of vocal volume. These anecdotal reports from participants and their families demonstrated that the SSP induces physiological changes and affects the sensitivity thresholds. In addition, the effect of the SSP varied from participant to participant, and some changes were too subtle to be reflected in a psychological test battery. Nevertheless, these changes were meaningful to each participant and his or her family, and most participants requested additional SSP trials. These improvements in ASD symptoms in their daily lives may be a possible reason for the partial association of the SRS-2 with rating scales that assess their quality of life, such as the WHOQOLBREF. 

This pilot study has several limitations. First, accuracy regarding statistical significance must be taken into account as this was a small exploratory study without a sham control group. Further large-scale studies are needed to confirm these effects and should include multicenter randomized controlled trials to minimize selection bias and placebo effects. Second, there were insufficient physiological and psychological assessments to evaluate the effects. There is a need for qualitative and quantitative studies that assess more objective and precise psychophysiological conditions, including sleep quality, traumatic experience, and behavioral state, in addition to self-reported measurements. Regarding behavioral changes, there is a limitation in assessing the behavior of adults and adolescents with ASD based on family reports. This is because the participants spend time alone in their room and family members may also be presenting with ASD. Therefore, there is a need for a tool that can quantitatively measure reciprocal communication. In addition, future studies should assess ANS function and its association with sleep, appetite, and psychological/behavioral states. Third, the follow-up period was only one month after the intervention. Given that subtle physiological and psychological changes may induce long-term behavioral changes, future studies should use a follow-up period of ≥3 months [40]. Fourth, in this experiment, the SSP was administered once daily (60 min) for five consecutive days. Each participant showed subtle changes, suggesting that adults and adolescents with ASD may benefit from long-term or repeated use of the SSP.

Addressing these limitations could allow the identification of cases that are most appropriate for SSP in terms of age, autism severity, type of comorbid psychiatric symptoms, and other comorbid developmental disorders. Above all, it is important for future research to create interventions with the input from autistic adults. In addition, this may provide clues for the development of novel treatments for adults and adolescents with ASD. 

## 5. Conclusions

Our findings suggest that the SSP has a partial effect on social impairments in adults with ASD. Specifically, we observed that the SSP led to an improvement in the social awareness subscale of SRS-2; furthermore, an effect on sensory communication skills can be expected. Although the SSP has limitations in its application to adults with ASD, it is a non-invasive and impressive short-term treatment program, and the fact that all participants were able to complete the entire program suggests that the SSP may become a viable approach to core symptoms of ASD. On the other hand, the SSP did not elicit active prosocial behavior as previously reported in children. This may be due to differences in neuroplasticity and flexibility; therefore, future studies in this area are needed. We hope that continuing to explore new approaches to ASD will assist them in living better lives in the future.

## Figures and Tables

**Figure 1 ijerph-20-04862-f001:**
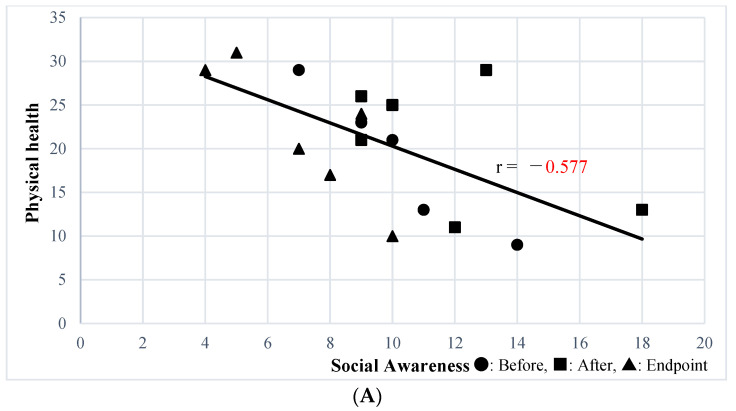
(**A**) Correlation between Social Awareness of the SRS-2 Family-Report and Physical Health of WHOQOL-BREF. (**B**) Correlation between Social Awareness of SRS-2 Family Report and the Anxiety State of STAI. (**C**) Correlation between Social Awareness of the SRS-2 Family-Report and the Anxiety Trait of STAI. (**D**) Correlation between Social Awareness of the SRS-2 Family-Report and CES-D.

**Table 1 ijerph-20-04862-t001:** Backgrounds of participants.

No.	Sex	Age	Occupation	History of Psychiatric Disorder	SRS2 Family-Report Evaluator
1	Male	21	student	adjustment disorder	mother
2	Male	24	unemployed	adjustment disorder	mother
3	Female	30	unemployed	adjustment disorder	mother
4	Male	23	unemployed	adjustment disorder	mother
5	Female	44	housewife	None	husband
6	Female	21	student	None	mother

**Table 2 ijerph-20-04862-t002:** Statistical analyses of SRS-2 Self-Report and Family Report.

SRS-2 Self-Report	Mean Value ± SD			df	*p* Value	
	Before	After	Endpoint		Before and After	Before and Endpoint
Social Awareness	12 ± 2.6	12.1 ± 5	13.6 ± 2.6	16	0.498	0.197
Social Cognition	19.1 ± 4.9	16.6 ± 7.3	19.5 ± 4.9	16	0.136	0.516
Social Communication	37.1 ± 9.4	36.1 ± 13.8	36.1 ± 8.4	16	1	0.753
Social Motivation	24.6 ± 4.7	23.8 ± 5.7	23.8 ± 4.9	16	0.236	0.496
Restricted Interests and Repetitive Behavior	18.8 ± 4.9	18.6 ± 8	19.1 ± 6.7	16	0.753	1
Total Score	111.8 ± 18.1	107.5 ± 36.7	112.3 ± 21.3	16	0.833	0.753
**SRS-2 Family Report**	**Mean Value ± SD**			**df**	***p*** **Value**	
	**Before**	**After**	**Endpoint**		**Before and After**	**Before and Endpoint**
Social Awareness	10.1 ± 2.3	11.8 ± 3.4	7.1 ± 2.3	16	0.596	0.027 *
Social Cognition	12 ± 4.2	14.8 ± 7	11.3 ± 5.6	16	0.131	0.414
Social Communication	22.1 ± 10.3	31.1 ± 14.1	21.8 ± 9.7	16	0.116	0.917
Social Motivation	14.1 ± 7	17.1 ± 5.7	15.6 ± 7.3	16	0.093	0.223
Restricted Interests and Repetitive Behavior	12.1 ± 6.8	15.1 ± 9.1	12.6 ± 7.2	16	0.144	0.414
Total Score	70.6 ± 22.5	90.1 ± 34.8	68.6 ± 24.6	16	0.116	0.786

* *p* < 0.05.

**Table 3 ijerph-20-04862-t003:** Correlations of subscales and total score of SRS-2 Self- or Family Report with the subscales of A/ASP.

SRS-2 Self-Report	Adolescent/Adult Sensory Profile			
	Low Registration	Sensation Seeking	Sensory Sensitivity	Sensation Avoiding
Social Awareness	0.347	−0.442	0.459	0.347
Social Cognition	0.587 **	−0.421	0.295	0.313
Social Communication	0.348	−0.396	−0.115	−0.0006
Social Motivation	0.589 *	−0.710 ***	0.318	0.562 *
Restricted Interests and Repetitive Behavior	0.696 **	−0.219	0.401	0.313
Total Score	0.487 *	−0.572 *	0.09	0.179
SRS-2 Family-Report	Adolescent/Adult Sensory Profile			
	Low Registration	Sensation Seeking	Sensory Sensitivity	Sensation Avoiding
Social Awareness	0.519 *	−0.212	0.368	0.399
Social Cognition	0.764 ***	−0.133	0.651 **	0.620 **
Social Communication	0.211	−0.191	−0.048	−0.07
Social Motivation	−0.021	−0.456	−0.107	−0.030
Restricted Interests and Repetitive Behavior	0.580 *	−0.223	0.442	0.350
Total Score	0.470 *	−0.107	0.220	0.157

* *p* < 0.05., ** *p* < 0.01., *** *p* < 0.001.

## Data Availability

The data presented in this study are available from the corresponding author (M.K.) on reasonable request.

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
