# Peer review of "Initial Outcomes of the Safe and Sound Protocol on Patients with Adult Autism Spectrum Disorder: Exploratory Pilot Study"

_ijerph, 2023, doi:10.3390/ijerph20064862_

Round 1

Reviewer 1 Report

A positive for this manuscript is that this is the first study to test the Safe and Sound protocol on adult ASD participants and the first independent lab (outside of the original authors) to attempt an evaluation. The paper is well-written and clearly lays out the need for empirical evaluation of this novel intervention. The authors acknowledge the limitations of the study such as the small sample size but need to emphasize the absence of a sham control. The one finding of improvement in a self-report measure might easily be explained by treatment bias. 

In the acknowledgements, the authors mention help with the heart rate variability (HRV) data. However, no such data was presented. This data, if available, would add value to the study. 

The authors chose to use nonparametric statistics, but failed to explain why. If the violations of normality or homogeneity were not severe, parametric statistics would offer greater power (though I doubt it would make much difference here). 

Overall, a useful first attempt to assess the Safe and Sound intervention. 

Author Response

Thank you for your comment. Please see the attachment.

Reviewer 2 Report

Thank you for offering me the opportunity to review this very interesting paper. In my opinion the paper provides novel findings that are important to be shared with the scientific community. However, some minor revisions are needed. Further clarifications are provided below:

Section 2.4 Assessment & measurements

·        The authors state that the SRS-2 has a conceptually derived “two-factor” structure, but then they report three different factors. Could the authors revise this accordingly?

·        There is no reference to the reliability and validity of the STAI, WHOQOL-BREF and A/ASP– could the authors add a sentence to them for reliability and validity as it was done for the SRS-2 and the CES-D?

·        In general the assessment section would benefit from reports on the reliability of the measures used in the described sample (i.e. calculations of the Cronbach’s alpha), although the sample is very small.

Section 2.5:

·        This section is also named as "Assessment & measurements". Was this intentional or do the authors want to name it differently to reflect its difference from Section 2.4?

·        What were the results of the normality test? It is a bit strange that the authors used a nonparametric test to analyze the data and then a parametric test for the correlations. Could the authors clarify and explain their choice?

Discussion:

·        The authors mention that there were age differences but there was no further explanation in the discussion how the outcomes might have differed for participant 5, who was significantly older than the other participants. In the 3rd paragraph of the discussion it seemed that participant 5 had better outcomes than the rest of the participants, although the authors suggest that in comparison to children, older participants were not as responsive to the treatment.

·        The discussion section might benefit from a reference to the fact that some differences and correlations might have been not significant because of the sample size. It is possible that the results were not significant because there was not enough power to find significant findings, so studies with more participants might be able to replicate these findings or indicate further improvements after treatment.

Author Response

(The authors gave the same response as above.)

Reviewer 3 Report

In general, I find this to be an interesting study. I do think that some references could be updated, as there is much newer work in the area of autism that should be cited (work in this area moves quickly, and our understanding of the disorder has shifted substantially).

I have a few concerns with the general framing of autism in this paper. For context, I am based in the United States. This paper uses deficit-based language throughout to describe members of the autism community (“impairment” and “dysfunctions” for example). This paper also uses the medical model to describe autism, essentially stating that characteristics of autism can be “treated.” In fact, that is the whole premise of the study. However, this is very problematic, as members of the autistic self advocacy community would argue that autism is not a disease, nor something to be treated. It is not something that goes away as one grows up, nor can it be “made better.” If anything, one can learn to hide characteristics, but even then, this is a terrible thing to ask of a person. It simply adds to their mental load. I would encourage the authors to explore the Autistic Self Advocacy Network (ASAN) for more information here. Perhaps this is different in other parts of the world, but in a journal that will reach many members of the English speaking autistic community, I believe there will be quite a bit of pushback on how this study is framed. Similarly, the paper uses person-first language which, again, is a concern for many autistic self advocates. I wonder if this study was developed with input from any autistic adults? This would help add to its credibility. I’m curious also as to the intellectual functioning of the participants, if this was a consideration. Were participants compensated? 5 sessions of an hour each is a substantial time commitment.  I wonder also about the placebo effect. As participants knew they were part of a trial, I wonder if this influenced their reporting of individual and minor gains – particularly as every participant noted gains in very different areas. In general, I find this to be an interesting, original paper with potential for replication. I do have some concerns with the framing of the study itself, but it may be that these concerns are unique to the context from which I come from. I would urge the authors to look into this before publishing, to ensure that they won’t receive major pushback from the autism community. Perhaps simply adding some context to the paper itself may help to alleviate some of these concerns (for example, instead of trying to “fix” or “treat” an autistic person, these efforts may help improve their lives).

Author Response

(The authors gave the same response as above.)

Round 2

Reviewer 3 Report

There are two small insertions that reference that research about autistic people should include them in the process, and that research should be done to improve their quality of life. I appreciate this.